# A Neuro-symbolic Approach to Inverse Design of Thin-layer Metamaterials Under Layout Constraints

## Abstract

Inverse design aims to compute physical structures that exhibit desired properties. A prominent application in Photonics is the inverse design of *metamaterials*, which are artificial composite structures created by stacking layers of different materials to achieve targeted optical responses. In addition to achieving the desired optical properties, designers often aim to ensure that the resulting metamaterials comply with specific *layout constraints*. Although many Deep Learning (DL) approaches have recently been proposed for inverse design, they generally fail to incorporate such constraints into the design process. In this paper, we propose a neuro-symbolic approach that combines DL-based inverse design methods with Semantic Loss to inject layout requirements into the inverse design process. Our experiments demonstrate that the proposed approach enables state-of-the-art inverse design techniques to comply with a variety of constraints inspired by the Photonics literature.

## 1 Introduction

Deep Learning (DL)-based approaches to inverse design allow for data-driven synthesis of physical systems. Ranging from molecules Yoo et al. (2023) and photonic devices to structures in automotive and aerospace applications Kim et al. (2022); Sun et al. (2015); Sekar et al. (2019), enabling designers and engineers to generate materials, tools, and molecules based on the properties they should exhibit. A particularly interesting example of inverse design arises in the domain of Photonics, where it has gained significant attention within the research community Kang et al. (2024); Jiang et al. (2020); Wiecha et al. (2021). In this field, a prominent application is the design of thin-layer *metamaterials*, which aims at devising artificial composite structures by stacking layers of different materials (see Figure 1) that feature some desired spectral responses.

In this context, the *forward problem* amounts to computing the optical response of a predefined structure. This task can be readily addressed by using established physics-based photonic simulators. On the other hand, the *inverse problem* amounts to devising novel structures' layouts that produce a specific desired response. Hence, in this latter case, one wants to estimate the thickness and the material to employ in each layer of the metamaterial to build. In general, the inverse problem remains fundamentally an ill-posed problem due to the non-uniqueness of the solution and the instability inherent in mapping responses back to structural parameters, which means that very different layouts in the design space can produce identical spectral outputs Zhang et al. (2018).

Recent literature widely acknowledges deep learning and generative methods as promising strategies to approach inverse design problems Park et al. (2024); Ren et al. (2020); Wiecha et al. (2021). Indeed, in the case where a dataset of couples design-response is available, such methods can learn complex, high-dimensional mappings that enable efficient exploration of the design space Park et al. (2024); Ren et al. (2020); Wiecha et al. (2021). Nonetheless, few approaches in the literature address the problem of *constrained* inverse design Piggott et al. (2017); Schubert et al. (2022). In this variant, the goal is not only to produce metamaterials that match a desired response, but also to ensure that they are physically realizable. This final requirement involves complying with domain-specific structural constraints, which requires integrating some expert knowledge into the design process. These constraints can serve multiple purposes, such as simplifying physical fabrication,

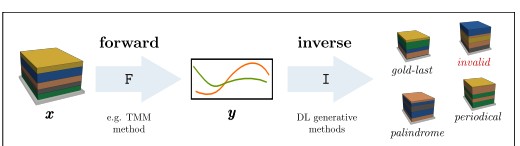

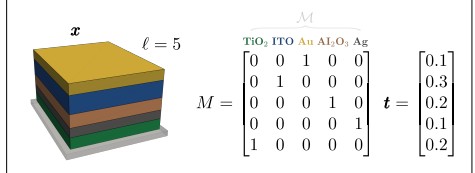

Figure 1: The *forward* and *inverse* problems. Generative methods produce valid or invalid metamaterials w.r.t. some structural layout constraints.

Figure 2: An example of a 5-layer metamaterial $\boldsymbol{x} = (M, \boldsymbol{t})$.

reducing costs, aligning with material availability in laboratory settings, incorporating domain expertise, and enforcing specific structural patterns For example, in the context of metamaterials design for biosensing, a common layout constraint might involve selecting gold (aurum) instead of silver for the outermost layer to prevent oxidation, while still achieving a desired optical response Joy et al. (2024). Another example concerns *hyperbolic metamaterials* Li & Gu (2024), where designers aim to produce artificial metamaterials by stacking thin-layers of different materials according to an ad-hoc defined repeated pattern Sreekanth et al. (2016). Such kinds of metamaterials and responses can be obtained by forcing *periodical* assignment to the layer. For instance, suppose we want to obtain a nine-layer metamaterial from a set of three materials $\{a, b, c\}$ of which one material is assigned to each layer. The design of a periodic (hyperbolic) metamaterial would produce a structure with the following layout: $(bac)(bac)(bac)$. Such structure might be required because it exhibits rare optical properties not found in natural materials, such as bending light in unconventional ways or letting it travel in directions that would normally be blocked Sreekanth et al. (2016). Currently available inverse design approaches *do not* support the incorporation of layout constraints of this kind, thus they limit the possibilities of designers and often lead to designs that are unfeasible or of limited practical relevance. In this work, we extend state-of-the-art inverse design approaches by incorporating domain knowledge into the design process, with the aim of producing feasible and domain-aware solutions for the inverse design of metamaterials. To this end, we propose the use of the neuro-symbolic *Semantic Loss* (SL) Xu et al. (2018) approach to introduce, into the DL-based inverse design models, a differentiable loss term that captures *how likely* a propositional theory (modeling domain constraints) is satisfied. In particular, we introduce SL into state-of-the-art inverse design processes and evaluate its effectiveness across a diverse set of layout constraints, such as those required to generate periodic metamaterials. Our experimental results show that:

    *i.* SL effectively enforces layout constraints on thin-layer metamaterials, leading to designs of practical and physical relevance.

    *ii.* Notably, we obtained the generation of valid metamaterials that are either underrepresented or entirely absent in the training data. Our models, hence, enforce constraints at inference time without requiring re-training or fine-tuning on datasets that explicitly include valid constrained examples.

    *iii.* The use of SL leads to designs that better align with desired properties, with respect to baseline approaches.

Our approach opens avenues to a new paradigm in inverse design, where designers can declaratively inject domain knowledge and layout constraints desiderata into the design process.

## 2 RELATED WORK

Deep learning-based inverse design architectures typically involve two main phases: a training phase of the different architectures' components (e.g. training $N_F$ on $\mathcal{D}$) and an inverse computation phase (e.g., prediction of the structure or its optimization). In Adornetto & Greco (2023), the authors propose a classification of DL-based inverse design approaches, distinguishing between *output-independent* methods which generate $\bar{\boldsymbol{x}}$ with simple inferences, and *output-dependent* methods, which use the desired output $\bar{\boldsymbol{y}}$ to explore the design space during inverse computation.

A key challenge in inverse design is ensuring that generated structures satisfy constraints. We distinguish between *fabrication-geometric* and *physical feasibility* constraints. The former arise from

manufacturing limitations, such as minimum feature sizes or permissible layer thicknesses, typically addressed by introducing penalty terms or filters and thresholding Vercruysse et al. (2019); Schubert et al. (2022); Vercruysse et al. (2019); Ma et al. (2024). In contrast, *physical feasibility constraints* require that designs obey the laws of physics and satisfy application-specific requirements. While physics laws (e.g., Maxwell's equations in optics) are typically enforced via surrogate models, additional constraints—such as symmetry, periodicity, or categorical representations—may also be necessary. Although well-known in photonics, these constraints are rarely addressed in inverse design frameworks, and typically implemented through penalty terms Adornetto & Greco (2023); Bastek et al. (2022); Jang et al. (2016); Ren et al. (2020) or introduced as hard constraints within constrained optimization solvers Lu et al. (2021).

This work focuses on feasibility constraints that enforce layouts in metamaterial design, where domain-specific structural rules can guide the design space exploration towards valid configurations. To the best of our knowledge, this paper proposes the first AI-based approach that leverages on neurosymbolic techniques to obtain an effective method for constrained inverse design of metamaterials.

## 3 PRELIMINARIES ON INVERSE DESIGN

### 3.1 METAMATERIALS

A *layered metamaterial* Lininger et al. (2021), in our context, refers to a structure composed of multiple metallic layers stacked vertically to achieve specific electromagnetic properties (see Figure 1).

More formally, let $\mathcal{M} = \{m_1, m_1, \ldots, m_q\}$ be a finite set of *materials*, $\ell$ a positive integer representing the number of layers in a metamaterial. An $\ell$-layer metamaterial over $\mathcal{M}$ is a tuple $\boldsymbol{x} = (M, \boldsymbol{t})$ where $M$ is a binary matrix of size $\ell \times q$, and the vector $\boldsymbol{t} = (t_1, \ldots, t_\ell)$ where $t_i \in \mathbb{R}^+$ is the thickness of the $i$-th layer of $\boldsymbol{x}$. We denote $M$ as the *material matrix* encoding the layout of the metamaterial, and $\mathbf{t}$ as the *thickness vector*. In this setting, the $i$-th row of $M$ is the one-hot encoding of a material such that $M_{i,j} = 1$ if and only if the material $m_j \in \mathcal{M}$ is assigned to the $i$-th layer of the metamaterial. Figure 2 provides a graphical representation of a metamaterial.

### 3.2 INVERSE DESIGN

In this work we consider inverse design problems with the availability of a dataset $\mathcal{D} = \{(\boldsymbol{x}_i, \boldsymbol{y}_i)\}_{i \in \{1, \ldots, n\}}$ of (structure-response) pairs such that, for each $i \in \{1, \ldots, n\}$, $\boldsymbol{x}_i$ is a metamaterial and $\boldsymbol{y}_i \in \mathbb{R}^m$ is a *spectral-response* vector. Typically, $\boldsymbol{y}_i$ represents a set of the metamaterial's optical properties (e.g., transmittance and reflectance) obtained through a real physical simulator $\mathtt{F}$ such that $\boldsymbol{y}_i = \mathtt{F}(\boldsymbol{x}_i)$ holds. The inverse design problem consists of computing, given a desired *target response* $\bar{\boldsymbol{y}}$, the input vector $\bar{\boldsymbol{x}}$ such that $\mathtt{F}(\bar{\boldsymbol{x}}) = \bar{\boldsymbol{y}}$. Informally, we are interested in learning the *inverse function* of $\mathtt{F}$, that is the function $\mathtt{F}^{-1}$ such that $\mathtt{F}^{-1}(y) = x$; however, it is often the case that $\mathtt{F}$ is not invertible, as different metamaterials can produce the same physical response. Hence, we can understand inverse design as the learning of a function $\mathtt{I}$ such that $\mathtt{F}(\mathtt{I}(\bar{\boldsymbol{y}})) \approx \bar{\boldsymbol{y}}$. Figure 1 depicts the classic inverse design framework.

Furthermore, it is often the case that physical simulators consist of iterative, non-differentiable algorithms. Thus, DL-based inverse design approaches rely on so called *surrogate models*, that is *differentiable approximations* of physical simulators, e.g., neural networks that approximate a simulator. Given a simulator $\mathtt{F}$, we use the notation $\mathtt{N_F}$ to denote its surrogate. In this setting, where neural networks are not inherently suited to produce one-hot encodings, we represent the generated design as $\bar{\boldsymbol{x}} = (\bar{M}, \boldsymbol{t}) \in \mathbb{R}^{\ell \times (q+1)}$, where $\bar{M}$ is a real-valued approximation of the binary material matrix $M$. The goal of the network is to guide $\bar{M}$ toward a one-hot (categorical) representation that aligns with the structure of valid metamaterials. This representation is required to ensure compatibility with the surrogate model $\mathtt{N_F}$, which is trained on datasets composed of well-represented metamaterial designs.

### 3.3 BENCHMARK MODELS

In this work, we focus on output-dependent methods, as they are better suited for integration with semantic loss during optimization and design space exploration. Accordingly, we selected the best-

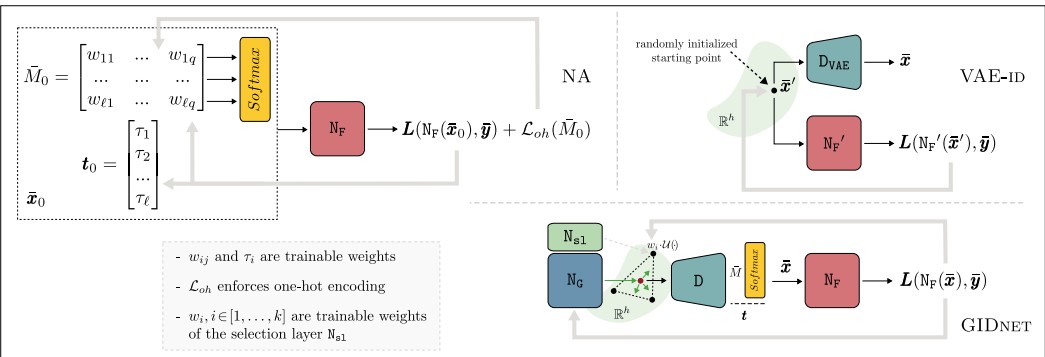

Figure 3: NA, VAE-ID and GIDNET frameworks for the metamaterial design.

performing state-of-the-art output-dependent architectures—proven to achieve superior results in the design of thin-layer metamaterials Adornetto & Greco (2023); Yang et al. (2023).

**Neural Adjoint (NA)**   It is part of a family of gradient-based inverse design methods Zaabab et al. (1995); Peurifoy et al. (2018); Asano & Noda (2018). They use the pre-trained network $\mathtt{N_F}$ (as a surrogate simulator), which is frozen during inverse computation phase. In such a phase, the network takes as input an initial randomly sampled guess $\bar{\boldsymbol{x}}_0$ of trainable weights. The loss function is meant to minimize the distance between $\bar{\boldsymbol{y}}$ and $\mathtt{N_F}(\bar{\boldsymbol{x}}_0)$, and it is optimized via backpropagation by directly updating the weights of $\bar{\boldsymbol{x}}_0$ (keeping $\mathtt{N_F}$ frozen). Eventually, the resulting design $\bar{\boldsymbol{x}}$ (such that $\mathtt{F}(\bar{\boldsymbol{x}}) \approx \bar{\boldsymbol{y}}$) is given by the values of the weights in $\bar{\boldsymbol{x}}_0$ after their optimization. A known limitation of this approach is that the search space defined by $\bar{\boldsymbol{x}}_0$ is often narrow, leading to suboptimal solutions Jiang et al. (2020) and hence, making NA particularly sensitive to the initialization of $\bar{\boldsymbol{x}}_0$. To mitigate this, Ren et al. (2020) proposes a resampling strategy where, for a given target response $\bar{\boldsymbol{y}}$, NA repeats $T$ times the optimization of $\bar{\boldsymbol{x}}_0$, starting from different random initializations of weights. In our experiments, we instantiate NA to work with a one-hot encoding loss $\mathcal{L}_{oh}$, specifically designed for metamaterial design as proposed in Adornetto & Greco (2023). This loss encourages valid material assignments by enforcing one-hot encodings across the rows of the real-valued $\bar{M}$. Figure 3 illustrates how we adapt the general NA framework to the metamaterial inverse design.

**VAE-ID**   Originally proposed for molecule inverse design in Gómez-Bombarelli et al. (2018), this architecture jointly trains a variational autoencoder Kingma & Welling (2013) and a variant $\mathtt{N_F}'$ of a surrogate model to guide optimization in a latent design space during the inverse computation. During training VAE-ID jointly optimizes a variational autoencoder and $\mathtt{N_F}'$ on both reconstruction and property prediction—making the latent design space a continuous representation of inputs conditioned by the responses Kingma & Welling (2013). During inverse computation phase VAE-ID initially samples a random point (or $T$ points in the case of resampling strategy) from the latent design space. Such starting point is then provided in input to the decoder of the variational autoencoder $\mathtt{D_{VAE}}$ to generate a candidate design $\bar{\boldsymbol{x}}$. The loss function is meant to minimize the distance between $\bar{\boldsymbol{y}}$ and $\mathtt{N_F}'(\bar{\boldsymbol{x}}')$. To this aim, both $\mathtt{D_{VAE}}$ and $\mathtt{N_F}'$ are frozen and the design is optimized by directly moving $\bar{\boldsymbol{x}}'$ into the latent space. Eventually the final latent design is decoded. Interestingly, no constraints are enforced during the exploration process, and the final design is validated ex-post.

**GIDNET**   It is a recently proposed approach to inverse design proposed in Adornetto & Greco (2023). During training phase, GIDNET constructs a latent space design with an autoencoder. The encoder $\mathtt{E}$ maps an input $\boldsymbol{x}$ to its latent representation $\boldsymbol{x}' \in \mathbb{R}^h$, while the decoder $\mathtt{D}$ attempts to reconstruct the original input. When required, the decoder is further trained to enforce categorical structure in the reconstructed design, by using a custom loss function $\mathcal{L}_{oh}$, which penalizes continuous outputs that deviate from a one-hot encoding. In the inverse phase, GIDNET 's *Selection Layer* $\mathtt{N_{sl}}$ first picks the $k$-nearest designs to the target response $\bar{\boldsymbol{y}}$ in the training set, then forms a trainable weighted combination of their latent representations, by identifying a suitable starting point. A generator $\mathtt{N_G}$ takes as input random noise and generates movements of the starting point in multiple directions within the latent space to produce a diverse set of candidates. These latent points

are decoded by D back into designs and evaluated with the surrogate $N_F$. The loss function aims to minimize the distance between the predicted response $N_F(D(\bar{\boldsymbol{x}}'))$ and the target response $\bar{\boldsymbol{y}}$, guiding the exploration. During this process, D and $N_F$ are kept frozen, while the parameters of the generator $N_G$ and the $k$ weights in $N_{s1}$ are updated to learn meaningful perturbations that improve design quality in the latent space. To ensure a fair comparison with other methods, such as NA and VAE-ID, which permit resampling of initialization points, we adapt GIDNET by modifying its initialization strategy. Specifically, instead of selecting the $k$ nearest neighbors to the target response $\bar{\boldsymbol{y}}$, we uniformly sample $k$ latent vectors within the bounds of the training set's distribution in the latent space. Such points are then combined to define a region of the latent space from which the exploration is initialized, as shown in Figure 3 (for $k = 3$).

## 4 CONSTRAINED INVERSE DESIGN BY SEMANTIC LOSS

In this section, we apply Semantic Loss Xu et al. (2018) (SL) to the inverse design problem. We start by recalling the definition of semantic loss:

**Definition 1** (Semantic Loss Xu et al. (2018)). *Let $\boldsymbol{X} = \{x_1, \ldots, x_n\}$ be a set of propositional variables, $\alpha$ a sentence over $\boldsymbol{X}$. Let $\boldsymbol{p}$ be a vector of probabilities, $\boldsymbol{p}_i$ being the probability associated to variable $x_i$. The semantic loss between $\alpha$ and $\boldsymbol{p}$ is:*

$$L^s(\alpha, \boldsymbol{p}) = -\log \sum_{x \models \alpha} \prod_{x_i \in x} p_i \prod_{x_i \notin x} 1 - p_i$$

Intuitively, $L^s(\alpha, \mathbf{p})$ penalizes probability distributions $\mathbf{p}$ from which it is "difficult to sample models of $\alpha$". This is formalised in Proposition 3 of Xu et al. (2018). Hence, semantic losses can be used as a loss term to push neural networks' outputs—in terms of $\mathbf{p}$—towards aligning semantically to the sentence $\alpha$. We refer the reader to Xu et al. (2018) for a thorough and formal treatment of semantic losses and their formal properties. In order to apply SL to the inverse design problem, we have thus to $(i)$ define an appropriate set of variables to *express design constraints* over metamaterials' structures and $(ii)$ interpret inverse design output as a probability distribution over such variables.

### 4.1 DEFINING DESIGN CONSTRAINTS

As the structure of a metamaterial $\boldsymbol{x} = (M, \boldsymbol{t})$ is described by its (binary) material matrix $M \in \{0, 1\}^{\ell \times q}$, it is naturally mapped to a collection of propositional variables (i.e., its non-zero entries), and $M$ can be understood as an interpretation for sentences (propositional logic formulae) over such variables. This makes propositional logic an adequate language to define (combinatorial) properties of materials, to be enforced at inference time by means of a semantic loss.

More formally, let $P = \{x^i_j : 1 < i \le \ell, 1 < j \le q\}$ be a set of propositional variables, with $x^i_j$'s truth value modeling that "the $i$-th layer of $x$ is composed by material $m_j$" (analogous to the interpretation of $M_{i,j} = 1$). Given a material matrix $M$, we define the two sets of variables $V^+(M) = \{x_{i,j} \in P : M_{i,j} = 1\}$ and $V^-(M) = P \setminus V^+(M)$.

Sentences over $P$, which we call *design constraints*, define *structural properties* for materials. Models of such formulae implicitly define a set of material matrices that satisfy it. We provide one example:

**Example 1.** *Assume we are interested in metamaterials where all adjacent layers use distinct materials. That is, whenever material $m_j$ appears in a layer $i$, we wish layer $i + 1$ to* not *be composed of material $m_j$. This can be expressed as the following propositional formula:*

$$\phi_{adj} = \bigwedge_{j \in \{1, \ldots, q\}} \bigwedge_{i \in \{1, \ldots, \ell-1\}} x^i_j \rightarrow \neg x^{i+1}_j$$

*By following our notation, the metamaterial shown in Figure 2 corresponds to the interpretation $V^+(M) = \{x^1_3, x^2_2, x^3_4, x^4_5, x^5_1\}$, which satisfies the formula $\phi_{adj}$. On the other hand, substituting the $m_3$ with $m_2$ in the first layer would yield the interpretation $\{x^1_2, x^2_2, x^3_4, x^4_5, x^5_1\}$, that does not satisfy $\phi_{adj}$ (layer 1 and layer 2 use both $m_2$).*

Similarly, a model $I \subseteq P$ of a design constraint $\phi$ can be mapped back onto a design matrix, *reshaping $I$* into the matrix $Mat(I)$ such that $Mat(I)_{i,j} = 1$ if and only if $x_{i,j} \in I$. Thus, with a slight abuse of notation, we say an inverse design $\bar{\boldsymbol{x}} = (M, \boldsymbol{t})$ *satisfies* the design constraint $\phi$ if $V^+(M)$ is a model of $\phi$, $M \models \phi$ in symbols. The *constrained* inverse design problem for the tuple $(\boldsymbol{y}, \phi)$ consists of computing $\bar{\boldsymbol{x}} = (M, \boldsymbol{t})$ such that $\mathtt{F}(\bar{\boldsymbol{x}}) = \boldsymbol{y}$ and $M$ satisfies $\phi$.

## 4.2 Assigning Probabilities to Design Constraints

Typically, deep learning-based inverse design methods for a target response $\bar{\boldsymbol{y}}$ are unable to *directly* output a binary matrix, but rather output (ignoring the thickness vector) a matrix of probabilities $\bar{M} \in [0, 1]^{q \times \ell}$, where the value $\bar{M}_{i,j} \in [0, 1]$ is to be interpreted as *the probability of layer $i$ being composed of material $m_j \in \mathcal{M}$*[1].

Then, a material matrix $M$ is obtained from $\bar{M}$ by setting $M_{i,j} = 1$ if and only if $m_j$ is the most probable material for layer $i$ according to $\bar{M}$. This suggests a natural way to apply SL to the problem of constrained inverse design. More in detail, for the constrained inverse design problem $(\bar{\boldsymbol{y}}, \phi)$ and a candidate solution (obtained by output-dependent inverse design methods) $\bar{\boldsymbol{x}} = (\bar{M}, \boldsymbol{t})$, the semantic loss $L^s(\phi, \cdot)$ is obtained by:

$$L^s(\phi, \bar{M}) = -\log \sum_{M \models \phi} \underbrace{\left( \prod_{x_j^i \in V^+(M)} \bar{M}_{i,j} \prod_{x_j^i \in V^-(M)} 1 - \bar{M}_{i,j} \right)}_{w(M)}$$

where the term $w(M)$ describes the probability associated to a specific model $M$ of the formula $\phi$[2]. Summing over all such $M$'s, we get the overall probability of satisfying $\phi$.

The approach herein proposed enables to declaratively write design constraints, expressed as propositional logic sentences, rather than devise ad-hoc loss terms as in available inverse design approaches. It also enables *compositionality*: two design constraints $\phi$ and $\varphi$ can be combined—searching for metamaterials that satisfy both—by considering their conjunction $\phi \wedge \varphi$; or, asserting that $\varphi$ must hold if $\phi$ does can be encoded as $\phi \rightarrow \varphi$, and so on.

Our approach can be integrated with any output-dependent inverse design method to impose constraints on the final layout. This holds whether the method operates directly in the original design space or within a latent representation. In both cases, the semantic loss guides the design—either directly or via its latent encoding—towards solutions that satisfy the constraints when eventually decoded back into the original space.

## 5 Experimental Setting

### 5.1 Datasets

In our experiments we use two state-of-the-art datasets for inverse design of metamaterials. Both the datasets consist of metamaterial-response pairs where each metamaterial $\boldsymbol{x}_i$ structure is associated with an optical response $\boldsymbol{y}_i$, obtained via the *transfer matrix method* (TMM) Chilwell & Hodgkinson (1984). However, they differ in the number of material layers, material choices, and the dimensionality of the optical response as shown in Table 1[3].
In our experiments on $\mathcal{D}_{\ell=5}$ and $\mathcal{D}_{\ell=10}$, we used 219500 and 44300 examples respectively, for pretraining the components, and 500 and 100 examples as test-set for inverse computation.

### 5.2 Layout Constraints

Our experiments feature the following design constraints:($i$) USEALL(UA), which forces the metamaterial to contain at least once each available material. ($ii$) NOADJACENTDUPLICATES(NAD), that forbids a material to appear in adjacent layers; ($iii$) PALINDROME(PAL$i$), that forces the design of

---

[1]This is easily achieved by means of a row-wise `softmax(·)`.
[2]SL assumes all entries of $\mathbf{p}$ are independent. Thus, $w(I)$ is the product of $\mathbf{p}_i$ for $x_i \in I$ and $1-\mathbf{p}_i$ for $x_i \notin I$.

Table 1: Datasets characteristics.

|  | $\ell$ | $|\mathcal{M}|$ | $\boldsymbol{x}$ | $\boldsymbol{y}$ | Source |
|---|---|---|---|---|---|
| $\mathcal{D}_{\ell=5}$ | 5 | 5 | $\{0,1\}^{5\times 5}\times\mathbb{R}^5$ | $\mathbb{R}^{2400}$ | (Lininger 2021) |
| $\mathcal{D}_{\ell=10}$ | 10 | 7 | $\{0,1\}^{10\times 7}\times\mathbb{R}^{10}$ | $\mathbb{R}^{2001}$ | (Yang 2023) |

Table 2: Percentage of constraints satisfaction.

| Constraint | $\mathcal{D}_{\ell=5}$ | | $\mathcal{D}_{\ell=10}$ | |
|---|---|---|---|---|
| | Train | Test | Train | Test |
| UA | 9.4% | 9.3% | 10.2% | 13.0% |
| NAD | 100% | 100% | 25.1% | 17.0% |
| PAL2 | 6.2% | 6.7% | 2.1% | 4.0% |
| PAL3 | – | – | 0.2% | 1.0% |
| PAL4 | – | – | 0% | 0% |
| P2 | 1.5% | 2% | 0% | 0% |
| P3 | 4.7% | 5.1% | 0% | 0% |
| P4 | – | – | 0% | 0% |

metamaterials where layers $i$ and $L-i$ are composed of the same material; $(iv)$ PERIODICAL(P$k$) that forces a structure where a metamaterial is present in a layer $i$ iff it appears also in the layer $i+k$, while not reappearing in every intermediate layer $i+1,\dots,i+k-1$. Furthermore, we always include a conjunction term in the semantic loss to avoid multiple materials being assigned to the same layer. Table 2 provides summary statistics about the amount of metamaterials in the datasets that comply with these constraints. Notice the constraints PAL3, PAL4, and P4 cannot be meaningfully enforced over $\mathcal{D}_{\ell=5}$, as there are not enough layers.

### 5.3 ARCHITECTURES INSTANTIATION

For our experiments, we instantiated three state-of-the-art output-dependent architectures: NA, VAE-ID, and GIDNET. We evaluated them on the two metamaterial datasets, comparing the performance of their original (*baseline*) implementations against their variant incorporating the $L^s$ term during inverse computation.

**Surrogate Models & Autoencoders**   All inverse design methods rely on a surrogate simulator to optimize candidate solutions during the inverse computation. We hence trained two surrogate, $\mathrm{N_F}$ and $\mathrm{N_F'}$, on each of the datasets $\mathcal{D}_{\ell=5}$, $\mathcal{D}_{\ell=10}$. The training objective was to minimize the mean squared error (MSE) between the predicted and the ground-truth spectral responses. The VAE-ID and GIDNET methods require a pretrained autoencoder (variational for VAE-ID) to reconstruct the latent representations of metamaterials. We employ the same Encoder–Decoder architecture for each method and experiment, by following the parametric configuration proposed in Adornetto & Greco (2023). The dimensionality of the latent space (defined as $\ell \times 3$) results in $h = 15$ for $\mathcal{D}_{\ell=5}$ and $h = 30$ for $\mathcal{D}_{\ell=10}$. Both the GIDNET autoencoder and VAE-ID variational autoencoder were trained to minimize the reconstruction error, using a composite loss function consisting of column-wise categorical cross-entropy for the materials matrix $M$ and MSE for the thickness vector $\boldsymbol{t}$. In the case of GIDNET, we additionally include the one-hot regularization term as originally proposed Adornetto & Greco (2023), while for VAE-ID, we incorporate the KL-divergence Kingma & Welling (2013). For fair comparison, we use the same surrogate model and autoencoder architectures for all the methods and experiments, while best hyperparameters configurations were selected through hyperparameter search [3]

**Inverse Computation**   During the inverse computation phase, all methods adopt a resampling strategy as in Ren et al. (2020). Specifically, for a given target $\bar{\boldsymbol{y}}$, we generate $T = 128$ initial random points; each point is optimized to generate a candidate final solution. The optimal learning rate for each of the three inverse design methods was determined by evaluating a randomly chosen subset of 10 samples from the test set across $lr \in \{0.01, 0.05, 0.1, 0.5\}$ and selecting the value that yielded the best performance for the corresponding method and dataset. GIDNET uses two additional components in inverse computation: a *Selection Layer* $\mathrm{N_{s1}}$ and a generator $\mathrm{N_G}$ to explore the latent space. Both components' configurations were taken from the original work Adornetto & Greco (2023). The dimensionality of $\mathrm{N_{s1}}$ is $k = 30$, hence each initialization point results from the weighted linear combination of 30 normally distributed samples. During this inverse computation phase, all relevant components—$\mathrm{N_F}$, $\mathrm{N_F'}$, $\mathrm{D_{VAE}}$, and $\mathrm{D}$—were kept frozen.

---

[3]See Appendix B–C for Detailed descriptions on data, hyperparameters, performance and timing.

Table 3: SRMSE (mean, variance) and percentage of valid designs for different constraints–architecture pairs. Best in bold. We report $n.d.$ if a method is unable to provide valid results for the given constraint; We report a dash (-) if the constraint cannot be enforced over a specific dataset.

| | | $\mathcal{D}_{\ell=5}$ | | | | | | $\mathcal{D}_{\ell=10}$ | | | | | |
| | | NA | | VAE-ID | | GIDNET | | NA | | VAE-ID | | GIDNET | |
| | | *baseline* | *with $L^s$* | *baseline* | *with $L^s$* | *baseline* | *with $L^s$* | *baseline* | *with $L^s$* | *baseline* | *with $L^s$* | *baseline* | *with $L^s$* |
| UA | SRMSE | $0.039\pm(0.05)$ | $\mathbf{0.019}\pm(0.02)$ | $0.228\pm(0.17)$ | $\mathbf{0.221}\pm(0.18)$ | $0.022\pm(0.03)$ | $\mathbf{0.019}\pm(0.03)$ | $0.088\pm(0.02)$ | $\mathbf{0.073}\pm(0.02)$ | $0.179\pm(0.04)$ | $\mathbf{0.175}\pm(0.04)$ | $0.080\pm(0.02)$ | $\mathbf{0.068}\pm(0.02)$ |
| | one-hot | $0.913\pm(0.12)$ | $\mathbf{0.961}\pm(0.08)$ | $0.912\pm(0.07)$ | $\mathbf{0.982}\pm(0.03)$ | $0.941\pm(0.05)$ | $\mathbf{0.978}\pm(0.05)$ | $0.980\pm(0.03)$ | $\mathbf{0.998}\pm(0.00)$ | $0.850\pm(0.05)$ | $\mathbf{0.962}\pm(0.02)$ | $0.846\pm(0.06)$ | $\mathbf{0.987}\pm(0.02)$ |
| | valid (%) | 99.80% | 100% | 100% | 100% | 95.40% | 100% | 100% | 100% | 100% | 100% | 83% | 100% |
| NA | SRMSE | $0.021\pm(0.04)$ | $\mathbf{0.016}\pm(0.02)$ | $0.240\pm(0.21)$ | $0.279\pm(0.22)$ | $\mathbf{0.011}\pm(0.01)$ | $\mathbf{0.011}\pm(0.01)$ | $0.081\pm(0.02)$ | $\mathbf{0.075}\pm(0.02)$ | $0.177\pm(0.05)$ | $\mathbf{0.180}\pm(0.04)$ | $0.072\pm(0.02)$ | $\mathbf{0.064}\pm(0.02)$ |
| | one-hot | $0.946\pm(0.08)$ | $\mathbf{0.966}\pm(0.08)$ | $0.943\pm(0.06)$ | $\mathbf{0.992}\pm(0.02)$ | $0.972\pm(0.04)$ | $\mathbf{0.988}\pm(0.03)$ | $0.984\pm(0.01)$ | $\mathbf{0.998}\pm(0.00)$ | $0.849\pm(0.06)$ | $\mathbf{0.963}\pm(0.02)$ | $0.855\pm(0.05)$ | $\mathbf{0.991}\pm(0.02)$ |
| | valid (%) | 100% | 100% | 100% | 100% | 100% | 100% | 100% | 100% | 100% | 100% | 100% | 100% |
| PAL2 | SRMSE | $0.086\pm(0.14)$ | $\mathbf{0.021}\pm(0.02)$ | $0.267\pm(0.23)$ | $0.306\pm(0.25)$ | $0.022\pm(0.02)$ | $\mathbf{0.016}\pm(0.02)$ | $0.112\pm(0.03)$ | $\mathbf{0.074}\pm(0.02)$ | $0.185\pm(0.04)$ | $\mathbf{0.178}\pm(0.04)$ | $0.087\pm(0.02)$ | $\mathbf{0.064}\pm(0.02)$ |
| | one-hot | $0.867\pm(0.17)$ | $\mathbf{0.947}\pm(0.11)$ | $0.926\pm(0.07)$ | $\mathbf{0.980}\pm(0.03)$ | $0.952\pm(0.05)$ | $\mathbf{0.986}\pm(0.02)$ | $0.950\pm(0.12)$ | $\mathbf{0.997}\pm(0.00)$ | $0.845\pm(0.06)$ | $\mathbf{0.955}\pm(0.02)$ | $0.851\pm(0.05)$ | $\mathbf{0.990}\pm(0.02)$ |
| | valid (%) | 98.00% | 100% | 100% | 100% | 97.20% | 100% | 95% | 100% | 91% | 100% | 86% | 100% |
| PAL3 | SRMSE | | | | | | | $0.134\pm(0.04)$ | $\mathbf{0.074}\pm(0.02)$ | $0.210\pm(0.06)$ | $\mathbf{0.191}\pm(0.06)$ | $0.090\pm(0.02)$ | $\mathbf{0.067}\pm(0.02)$ |
| | one-hot | – | – | – | – | – | – | $0.825\pm(0.22)$ | $\mathbf{0.997}\pm(0.00)$ | $0.833\pm(0.06)$ | $\mathbf{0.953}\pm(0.02)$ | $0.841\pm(0.06)$ | $\mathbf{0.987}\pm(0.02)$ |
| | valid (%) | | | | | | | 28% | 100% | 33% | 100% | 23% | 100% |
| PAL4 | SRMSE | | | | | | | $0.121\pm(0.04)$ | $\mathbf{0.076}\pm(0.02)$ | $0.251\pm(0.04)$ | $\mathbf{0.203}\pm(0.06)$ | $0.103\pm(0.03)$ | $\mathbf{0.074}\pm(0.02)$ |
| | one-hot | – | – | – | – | – | – | $0.782\pm(0.30)$ | $\mathbf{0.996}\pm(0.00)$ | $0.819\pm(0.07)$ | $\mathbf{0.943}\pm(0.03)$ | $0.840\pm(0.06)$ | $\mathbf{0.986}\pm(0.02)$ |
| | valid (%) | | | | | | | 5% | 100% | 7% | 100% | 3% | 100% |
| P2 | SRMSE | $0.128\pm(0.16)$ | $\mathbf{0.024}\pm(0.03)$ | $0.338\pm(0.25)$ | $\mathbf{0.330}\pm(0.26)$ | $\mathbf{0.029}\pm(0.03)$ | $0.027\pm(0.04)$ | $n.d.$ | $\mathbf{0.110}\pm(0.02)$ | $n.d.$ | $\mathbf{0.264}\pm(0.07)$ | $n.d.$ | $\mathbf{0.147}\pm(0.06)$ |
| | one-hot | $0.738\pm(0.25)$ | $\mathbf{0.908}\pm(0.16)$ | $0.913\pm(0.07)$ | $\mathbf{0.969}\pm(0.05)$ | $0.936\pm(0.06)$ | $\mathbf{0.978}\pm(0.05)$ | $n.d.$ | $\mathbf{0.953}\pm(0.13)$ | $n.d.$ | $\mathbf{0.880}\pm(0.06)$ | $n.d.$ | $\mathbf{0.933}\pm(0.05)$ |
| | valid (%) | 50.20% | 100% | 82.00% | 100% | 61.20% | 100% | 0% | 100% | 0% | 34% | 0% | 100% |
| P3 | SRMSE | $0.080\pm(0.12)$ | $\mathbf{0.019}\pm(0.02)$ | $0.268\pm(0.22)$ | $0.284\pm(0.23)$ | $0.026\pm(0.03)$ | $\mathbf{0.016}\pm(0.02)$ | $n.d.$ | $\mathbf{0.087}\pm(0.02)$ | $n.d.$ | $\mathbf{0.217}\pm(0.06)$ | $n.d.$ | $\mathbf{0.097}\pm(0.03)$ |
| | one-hot | $0.854\pm(0.18)$ | $\mathbf{0.949}\pm(0.11)$ | $0.918\pm(0.07)$ | $\mathbf{0.978}\pm(0.04)$ | $0.948\pm(0.05)$ | $\mathbf{0.988}\pm(0.03)$ | $n.d.$ | $\mathbf{0.989}\pm(0.04)$ | $n.d.$ | $\mathbf{0.897}\pm(0.06)$ | $n.d.$ | $\mathbf{0.962}\pm(0.04)$ |
| | valid (%) | 91.80% | 100% | 98.60% | 100% | 84.00% | 100% | 0% | 100% | 0% | 75% | 0% | 100% |
| P4 | SRMSE | | | | | | | $n.d.$ | $\mathbf{0.082}\pm(0.02)$ | $n.d.$ | $\mathbf{0.203}\pm(0.05)$ | $n.d.$ | $\mathbf{0.082}\pm(0.03)$ |
| | one-hot | – | – | – | – | – | – | $n.d.$ | $\mathbf{0.995}\pm(0.00)$ | $n.d.$ | $\mathbf{0.917}\pm(0.05)$ | $n.d.$ | $\mathbf{0.976}\pm(0.03)$ |
| | valid (%) | | | | | | | 0% | 100% | 0% | 93% | 0% | 100% |

## 6 RESULTS

**Metrics** Performances of the methods have been compared via the spectral root mean squared error (SRMSE) as defined in Lininger et al. (2021); Adornetto & Greco (2023) between the spectral response associated with the metamaterial designed by the methods and the actual ones. The spectral response of the designed metamaterial was computed by using the real simulator F, hence comparing, for a generic inverse design method I, the responses $\texttt{F}(\texttt{I}(\bar{\boldsymbol{y}}_i))$ and $\boldsymbol{y}_i$ for each $\boldsymbol{y}_i$ in the test set. For all the methods we choose the best design over the optimization epochs—not necessarily taken from the last epoch—in the inverse computation phase (the one associated to the lower SRMSE between $\texttt{N}_\texttt{F}(\texttt{I}(\bar{\boldsymbol{y}}_i))$ and $\boldsymbol{y}_i$) out of the $T = 128$ optimized starting designs. Moreover, on the best selected design, we evaluate the one-hot metric as defined in Adornetto & Greco (2023). Finally, we use the valid (%) percentage to measure the satisfaction of constraints. In particular, for each sample in the test set we attempt T times the inverse design according to a constraint $\phi$. Let $p$ be the fraction of valid designs over T. We are interested in the average of $p$ over all metamaterials in the test set.

**The effect of SL** Table 3 reports results of our experiments across all architectures (baselines and semantic loss-augmented versions) and constraints of Section 4.3, for the two datasets $\mathcal{D}_{\ell=5}$ and $\mathcal{D}_{\ell=10}$. For each configuration, we perform $e = 200$ inverse design iterations, from $T = 128$ distinct starting points for each material. Table cells report mean and variance of the *best* SRMSE found (that is, over the $e \cdot T = 128 \cdot 200$ candidate solutions for each material), as well as the percentage of materials satisfying the constraints. Usage of SL always improves wrt the baseline architectures. This is observed both in terms of best SRMSE and percentage of materials satisfying the constraints. In Table 3 this effect is more evident for $\mathcal{D}_{\ell=10}$ dataset, where inverse designs complying with constraints PAL3, PAL4, P2, P3, P4 are found only by semantic loss-augmented architectures. On the other hand, the constraints UA, NA, PAL2 can also be solved by baseline architectures, but the semantic loss-augmented architectures achieve a lower SRMSE. Notably, the NA and GIDNET augmented with SL are able to provide valid designs for all materials in the datasets, for all constraints. We recall that the 100% in table refer to the number of valid materials. Output-dependent inverse design methods, like the ones we consider, typically optimize each material individually, so it's expected that outputs satisfy the design constraints when they are involved in the optimization. While a reader might associate this with overfitting, this is not the case since the goal is precisely to find a valid solution for each target. Generalization is not required for the inverse design process itself but may be relevant for the surrogate models, whose performances are reported in the Appendix.

**Comparing inverse designs across iterations** We analyze how the materials generated during the inverse design process differ between the baseline architecture and the version augmented with semantic loss. We report our results for the PAL3 constraint $\mathcal{D}_{\ell=10}$ [4].

---

[4]Appendix D includes similar analyses for the other architecture–constraint pairs.

Figure 4 shows a scatterplot, where a point $(x, y)$ denotes that on a given material, a baseline technique achieves a best SRMSE of $x$ while the semantic augmented version achieves a best SRMSE of $y$ across $e$ attempts and $T$ inverse design steps. Thus, intuitively, points that lie *below the bisector* in the scatterplot represent materials where the augmented architecture improves wrt the baseline SRMSE. Moreover, the color provides information about which methods were able to find inverse designs complying with the constraints. Grey points represent materials that both baseline and semantic loss-augmented architectures can design by satisfying the constraint. Similarly, materials where inverse design is not successful, with or without semantic loss, are in black. Conversely, green points are materials that can be successfully inverse designed solely by the semantic loss-augmented architecture, and red points are materials successfully designed solely by the baseline architecture.

First, we observe that all architectures are successful in jointly optimizing SRMSE and semantic loss, without drastically affecting SRMSE, as most points achieve low SRMSE. Explicitly optimizing the semantic loss yields a greater percentage of valid designs wrt the baseline (that is, *green points*). SL-augmented architectures were able to find some valid designs with a single optimization epoch, while baseline architectures required higher $T$ to find valid designs. In this regard, the NA and GIDNet architectures are more effective at $T = 1$. As $T$ and $e$ increase baseline architectures "naturally stumble upon" valid inverse designs (e.g., green points become grey); however, all architectures benefit from semantic loss achieving 100% of valid metamaterials (gray and green points) and overall lower SRMSE than their baseline counterpart, as we can observe from the mass of gray points below the bisector.

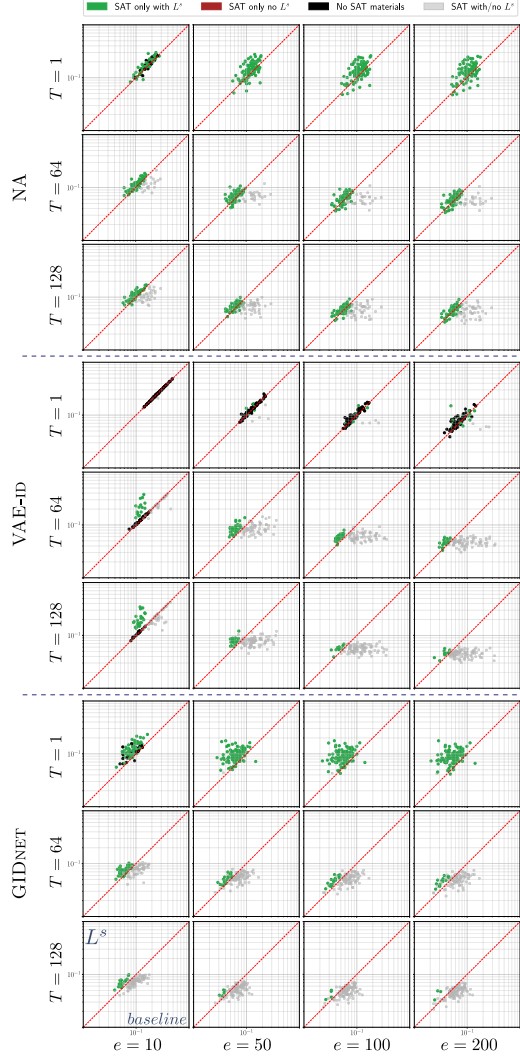

Figure 4: Effect of increasing inverse design iterations $T$ and using multiple starting points $e$ in the NA, VAE-ID, GIDNET architectures, over the PAL3 layout constraint in $\mathcal{D}_{\ell=10}$.

## 7 CONCLUSION

In this paper, we propose the application of Semantic Loss to solve constrained inverse design problems. Currently available inverse design approaches *do not* support this use case, which is essential to guarantee practical relevance of the proposed designs. We implement our approach on-top of state-of-the-art output-dependent inverse design methods. Experiments show our approach is effective at enforcing layout constraints on metamaterial designs, increasing the percentage of valid designs as well as lowering the error on the desired properties. Another advantage is that the approach does not require re-training, fine-tuning surrogate models (or other neural networks), nor augmenting datasets with examples that show the desired properties. Semantic losses can be chosen on a material-by-material basis, without affecting the overall architectures. As future works, we plan to explore richer formalisms to express design constraints, such as DeepProblog Manhaeve et al. (2021). This could allow to define constraints in a more natural fashion wrt SAT, relying on the same technical means (e.g., knowledge compilation) to achieve differentiability.

## REPRODUCIBILITY STATEMENT

All experiments presented in this work are fully reproducible. To ensure this, we provide detailed information on experimental settings in the Appendix, along with the complete source code as supplementary material concerning the experimental environment, dataset generation procedures, and all configuration parameters (including random seed values). These resources are designed to allow for an exact replication of the experiments and results. Additionally, we include a user guide that explains how to set up the environment and rerun the experiments step by step.

## ETHICS STATEMENT

This work does not involve human participants, animals, sensitive data, or any procedures that raise ethical concerns. No ethical issues are associated with the research presented in this paper.

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

APPENDIX

# A BENCHMARK MODELS

## A.1 NEURAL ADJOINT (NA)

It is part of a family of gradient-based inverse design methods Zaabab et al. (1995); Peurifoy et al. (2018); Asano & Noda (2018). They use the pre-trained network $\mathtt{N_F}$ (as a surrogate simulator), which is frozen during the inverse computation phase. In such a phase, the network takes as input an initial randomly sampled guess $\bar{\boldsymbol{x}}_0$ of trainable weights. The loss function is meant to minimize the distance between $\bar{\boldsymbol{y}}$ and $\mathtt{N_F}(\bar{\boldsymbol{x}}_0)$, and it is optimized via backpropagation by directly updating the weights of $\bar{\boldsymbol{x}}_0$ (keeping $\mathtt{N_F}$ frozen). Eventually, the resulting design $\bar{\boldsymbol{x}}$ (such that $\mathtt{F}(\bar{\boldsymbol{x}}) \approx \bar{\boldsymbol{y}}$) is given by the values of the weights in $\bar{\boldsymbol{x}}_0$ after their optimization. A known limitation of this approach is that the search space defined by $\bar{\boldsymbol{x}}_0$ is often narrow, leading to convergence to suboptimal local minima Jiang et al. (2020) and hence, making NA particularly sensitive to the initialization of $\bar{\boldsymbol{x}}_0$. To mitigate this, Ren et al. (2020) proposes an extension that resamples the $\bar{\boldsymbol{x}}_0$ multiple times. For a given target response $\bar{\boldsymbol{y}}$, NA repeats $T$ times the optimization of $\bar{\boldsymbol{x}}_0$, starting from different random initializations of its weights. In addition, Ren et al. (2020) introduces a *boundary loss* $\mathcal{L}_{bnd}$ to constrain the final design $\bar{\boldsymbol{x}}$ to be a normally distributed variable. However, this constraint is tailored for real-valued design spaces and does not directly suit metamaterials, where the design involves a binary layout matrix $M$ representing material assignments to layers. For this reason, in our experiments, we replace $\mathcal{L}_{bnd}$ with a one-hot encoding loss $\mathcal{L}_{oh}$, specifically designed for metamaterial design as proposed in Adornetto & Greco (2023). This loss encourages valid material assignments by enforcing one-hot encodings across the rows of the real-valued $\tilde{M}$. Figure 3 in the main paper illustrates how we adapt the general NA framework to the metamaterial inverse design in our experimental setting.

## A.2 VAE-ID

Originally proposed for molecule inverse design in Gómez-Bombarelli et al. (2018), this architecture jointly trains a variational autoencoder Kingma & Welling (2013) and a variant $\mathtt{N_F}'$ of a surrogate model to guide optimization during the inverse computation. In particular, the variational autoencoder is an encoder–decoder pair $(\mathtt{E_{VAE}}, \mathtt{D_{VAE}})$. The encoder $\mathtt{E_{VAE}}$ maps an input $\boldsymbol{x}$ to the parameters of a multivariate Gaussian distribution, defining a continuous latent representation of dimension $h$. A latent vector $\boldsymbol{x}' \in \mathbb{R}^h$ is then sampled from this distribution, typically using the reparameterization trick Kingma & Welling (2013). The decoder $\mathtt{D_{VAE}} : \mathbb{R}^h \to \mathbb{R}^{\ell \times (q+1)}$ generates a candidate design from $\boldsymbol{x}'$. On the other hand, $\mathtt{N_F}' : \mathbb{R}^h \to \mathbb{R}^m$ maps the latent representation of the input $\boldsymbol{x}'$ to the property $\boldsymbol{y}$. During training VAE-ID aims to jointly optimize $\mathtt{E_{VAE}}$, $\mathtt{D_{VAE}}$, and $\mathtt{N_F}'$ on reconstruction and property prediction—to let the latent space be a continuous representation conditioned by the properties.

By leveraging such representation, in the inverse computation phase VAE-ID initially samples a random point (or $T$ points in the case of resampling strategy) from the latent design space. Such starting point is then provided in input to $\mathtt{D_{VAE}}$ to generates a candidate design $\bar{\boldsymbol{x}}$. The loss function is meant to minimize the distance between $\bar{\boldsymbol{y}}$ and $\mathtt{N_F}'(\bar{\boldsymbol{x}}')$. To this aim, both $\mathtt{D_{VAE}}$ and $\mathtt{N_F}'$ are frozen, and the design is optimized by directly moving $\bar{\boldsymbol{x}}'$ to explore the latent space. At the end of this optimization, the final latent design is eventually decoded. Interestingly, no constraints are enforced during the exploration process; instead, the validity of the final designs is assessed only ex-post.

In the application to metamaterials design, no modification is needed to enforce onehot encoding of the generated metamaterial. Indeed, while in the other architectures, this is done to let suitably defined inputs to $\mathtt{N_F}$, in this case, $\mathtt{N_F}'$ is trained to work with continuous representations coming from the latent design space.

## A.3 GIDNET

It is a recently proposed approach to inverse design proposed in Adornetto & Greco (2023). During the training phase, GIDNET constructs a latent space using an autoencoder composed of an encoder–decoder pair $(\mathtt{E}, \mathtt{D})$. The encoder $\mathtt{E}$ maps an input $\boldsymbol{x}$ to its latent representation $\boldsymbol{x}' \in \mathbb{R}^h$,

while the decoder D attempts to reconstruct the original input. When required, the decoder is further trained to enforce categorical structure in the reconstructed design. To this end, the authors introduce a custom loss function $\mathcal{L}_{oh}$, which penalizes continuous outputs that deviate from a one-hot encoding.

During the inverse computation phase, GIDNET employs a dedicated mechanism known as *Selection Layer* to identify a suitable region of the latent space to explore. This is achieved by selecting a set of $k$ candidate designs—typically the $k$ nearest neighbors to the target response $\bar{y}$ in the training dataset—and computing a linear combination of their latent representations. Each candidate is weighted by a trainable parameter in the layer $\mathtt{N_{s1}}$, allowing the model to flexibly explore the latent space around a meaningful region. From this initialization point, normally distributed random noise is added and passed through the generator $\mathtt{N_G}$, which perturbs the point in multiple directions within the latent space to produce a diverse set of candidate solutions. These latent candidates are then decoded via D into the original design space and subsequently evaluated by the surrogate model $\mathtt{N_F}$. The loss function—designed to minimize the distance between the predicted response $\mathtt{N_F}(\bar{x})$ and the target response $\bar{y}$, guiding the exploration. During this process, D and $\mathtt{N_F}$ are kept frozen, while the parameters of the generator $\mathtt{N_G}$ and the $k$ weights in $\mathtt{N_{s1}}$ are updated to learn meaningful perturbations that improve design quality in the latent space.

To ensure a fair comparison with other methods, such as NA and VAE-ID, which permit resampling of initialization points, we adapt GIDNET by modifying its initialization strategy. Specifically, instead of selecting the $k$ nearest neighbors to the target response $\bar{y}$, we uniformly sample $k$ latent vectors within the bounds of the training set's distribution in the latent space. Such points are then combined to define a region of the latent space from which the exploration is initialized, as shown (for $k = 3$) in Figure 3 of the main paper.

Notably, GIDNET has demonstrated superior performance in several real-valued inverse design problems, as well as in the inverse design of metamaterials, making it the state-of-the-art architecture in this domain Adornetto & Greco (2023).

# B EXPERIMENTAL SETTING

## B.1 DATASETS

In our experiments, we use two state-of-the-art datasets for the inverse design of metamaterials. Both datasets consist of metamaterial-response pairs where each metamaterial $x_i$ structure is associated with an optical response $y_i$, obtained via the *transfer matrix method* (TMM) Chilwell & Hodgkinson (1984). However, they differ in the number of material layers, and the dimensionality of the optical response:

- $\mathcal{D}_{\ell=5}$ proposed in Lininger et al. (2021), in this dataset structures are made of 5 layers, and the materials set of 5 choices is defined as $\mathcal{M} = \{Ag, Al2O3, ITO, Ni, TiO2\}$. Each layer thickness is defined in the range $[1, 60]$nm. The input space is therefore $\mathbb{R}^{5 \times (5+1)}$. Each structure is associated with reflectance and transmittance spectra, for different polarizations, incident angles for 200 equally spaced points over the range $[450, 950]$nm (with values in $[0, 1]$). The output space is $\mathbb{R}^{2 \times 2 \times 3 \times 200}$. In our experiments on this dataset, we used 219500 examples as training-set and 500 examples as test-set.

- $\mathcal{D}_{\ell=10}$: proposed in Yang et al. (2023), in this dataset structures are made of 10 layers[5], and the materials set of 7 choices[6] is defined as $\mathcal{M} = \{ZnO, AlN, Al_2O_3, MgF_2, SiO_2, TiO_2, SiC\}$. Each layer thickness is defined in the range $[0, 1]$. The input space is therefore $\mathbb{R}^{10 \times (7+1)}$. The response y is a 2001-dimensional real-valued vector representing the average spectral reflectivity averaged over two polarizations, for different incident angles across 2001 equally-spaced wavelengths, in range $[0.3, 20]\mu$m. In our experiments, we used 44300 examples of the dataset as training-set and 100 examples as test-set.

---

[5]Technically, these metamaterials consist of 11 layers, but the final layer is always $Ag$ with a thickness of 0.1 $\mu$m and is not part of the design space.

[6]$Ag$ appears exclusively in the final layer, hence it is excluded from the set of available materials.

## B.2 Architectures Instantiation

For our experiments, we instantiated three state-of-the-art output-dependent architectures: NA, VAE-ID, and GIDNET. These architectures were evaluated on the two metamaterial datasets for the inverse design task, comparing the performance of their original (*baseline*) implementations against their variant incorporating the $L^s$ term during the inverse computation phase. Architectures' components instantiations follow.

**Surrogate models** All inverse design methods considered rely on a surrogate simulator model to evaluate and optimize candidate solutions during the inverse computation phase. We hence trained two surrogate, $N_F$ and $N_F'$, on each of the datasets $\mathcal{D}_{\ell=5}$, $\mathcal{D}_{\ell=10}$. The training objective for all the surrogates was to minimize the mean squared error loss function (MSE) between the predicted and the ground-truth spectral responses. To ensure a fair comparison across methods, we use the same surrogate model architecture—with the same number of network parameters—for all the experiments. For $\mathcal{D}_{\ell=5}$, we adopt the same $N_F$ architecture—matching the number and configuration of neural network layers—used in Adornetto & Greco (2023). The only exception is for VAE-ID, which operates in a latent space; in this case, the input layer of $N_F'$ is adjusted to match the latent dimensionality $h$. Model selection for $N_F$ was performed on $\mathcal{D}_{\ell=5}$ via grid search over the following hyperparameter space: learning rate $lr \in \{0.001, 0.005, 0.01, 0.05\}$, number of training epochs $e \in \{50, 100, 150, 200\}$, and batch size $bs \in \{256, 512, 1024\}$. The goal was to identify the configuration yielding the best predictive performance, measured in terms of MSE. The best-performing configuration was $lr = 0.005$, $e = 150$, and $bs = 1024$. The hyperparameter search was therefore aimed at identifying the configuration that achieved the lowest predictive MSE. The best-performing configuration was $lr = 0.005$, $e = 150$, and $bs = 1024$. A learning rate scheduling strategy (ReduceLROnPlateau) with a patience of 10 epochs was applied in all training runs. The final $N_F$ models trained on $\mathcal{D}_{\ell=5}$ achieved an MSE of 0.0003 on the test set, with spectral responses $y_i$ normalized to the range $[0, 1]$.

For $\mathcal{D}_{\ell=10}$ we configured $N_F$ as feed forward neural network of 3 fully connected subsequent layers of 80, 420, 640, 2001, 2001 neurons respectively. Again for VAE-ID, the input layer of $N_F'$ is adjusted to match the latent dimensionality $h$. Model selection for $N_F$ was performed on $\mathcal{D}_{\ell=10}$ via grid search over the same hyperparameter space (and ReduceLROnPlateau strategy) defined above for $\ell = 5$. The configuration yielding the best MSE was $lr = 0.005$, $e = 200$, $bs = 256$. The final $N_F$ models trained on $\mathcal{D}_{\ell=10}$ achieved an MSE of 0.0028 on the test set, with spectral responses $y_i$ originally defined in the range $[0, 1]$.

**Autoencoders** The VAE-ID and GIDNET methods require a pretrained autoencoder (variational in the case of VAE-ID) to reconstruct the latent representations of metamaterials. To ensure a fair comparison across methods, we employ the same Encoder–Decoder architecture—with an identical number of network parameters—for each experiment on dataset $\mathcal{D}_{\ell=i}$, where $i \in 5, 10$. For both datasets, the architecture follows the parametric configuration proposed in Adornetto & Greco (2023). According to this configuration, the dimensionality of the latent space (defined as $\ell \times 3$) results in $h = 15$ for $\mathcal{D}_{\ell=5}$ and $h = 30$ for $\mathcal{D}_{\ell=10}$. Since VAE-ID relies on a variational autoencoder, its architecture was modified to include two additional linear layers that map the Encoder's output to the mean and log-variance parameters of the latent Gaussian distribution, following the original formulation of the variational autoencoder Kingma & Welling (2013). Both the GIDNET autoencoder and VAE-ID variational autoencoder were trained to minimize the reconstruction error, using a composite loss function consisting of column-wise categorical cross-entropy for the materials matrix $M$ and MSE for the thickness vector $\boldsymbol{t}$. In the case of GIDNET, we additionally include the one-hot regularization term introduced by the authors in Adornetto & Greco (2023), while for VAE-ID, we incorporate the Kullback–Leibler divergence term as defined in the original variational framework. For both datasets and methods, we performed grid-search on a hyperparameter space defined by: $lr \in \{0.001, 0.005, 0.01, 0.05\}$, $e \in \{50, 100, 150\}$), and $bs \in \{128, 256, 512, 1024\}$.

For the autoencoder on $\mathcal{D}_{\ell=5}$, the best-performing configuration was $lr = 0.001$, $e = 150$, and $bs = 1024$, achieving a material assignment accuracy of 1.000 (i.e., the average proportion of correctly assigned materials per layer) and a thickness reconstruction MSE of $1.41 \times 10^{-4}$ on the test set. For the variational autoencoder on $\mathcal{D}_{\ell=5}$, the optimal configuration was $lr = 0.001$, $e = 150$, and $bs = 256$, with a reconstruction MSE of 0.035. On $\mathcal{D}_{\ell=10}$, the best AE configuration remained the same ($lr = 0.001$, $e = 150$, $bs = 1024$), achieving an accuracy of 1 and a reconstruction MSE

Table 4: Execution times (in seconds) for the inverse computation of a single metamaterial, $e = 200$ and $T = 1$

| | $\mathcal{D}_{\ell=5}$ | | | | | | $\mathcal{D}_{\ell=10}$ | | | | | |
| | NA | | VAE-ID | | GIDNET | | NA | | VAE-ID | | GIDNET | |
| | $baseline$ | $with\ L^s$ | $baseline$ | $with\ L^s$ | $baseline$ | $with\ L^s$ | $baseline$ | $with\ L^s$ | $baseline$ | $with\ L^s$ | $baseline$ | $with\ L^s$ |
|---|---|---|---|---|---|---|---|---|---|---|---|---|
| UA | 3.42 | 15.97 | 3.05 | 13.93 | 6.02 | 16.65 | 1.38 | 261.85 | 1.17 | 245.37 | 4.01 | 263.23 |
| NA | 3.68 | 14.28 | 3.98 | 14.47 | 6.03 | 14.06 | 1.15 | 45.42 | 1.23 | 44.04 | 3.63 | 46.40 |
| PAL2 | 3.37 | 14.98 | 3.29 | 14.29 | 5.64 | 16.27 | 1.24 | 26.43 | 1.13 | 27.06 | 3.56 | 27.08 |
| PAL3 | – | – | – | – | – | – | 1.23 | 55.95 | 1.13 | 56.53 | 3.57 | 58.45 |
| PAL4 | – | – | – | – | – | – | 1.24 | 186.59 | 1.13 | 187.61 | 3.57 | 204.47 |
| P2 | 3.44 | 13.53 | 3.38 | 13.67 | 5.63 | 12.72 | 1.24 | 34.34 | 1.14 | 35.35 | 3.61 | 35.83 |
| P3 | 3.14 | 13.93 | 3.27 | 13.91 | 5.27 | 14.37 | 1.21 | 65.09 | 1.12 | 65.68 | 3.57 | 68.33 |
| P4 | – | – | – | – | – | – | 1.24 | 109.33 | 1.13 | 110.18 | 3.59 | 113.00 |

of $1.08 \times 10^{-3}$ on the test set. For the VAE on $\mathcal{D}_{\ell=10}$, the best setup was $lr = 0.001$, $e = 100$, and $bs = 256$, resulting in a reconstruction MSE of $4 \times 10^{-4}$ on the test set.

**Other components** GIDNET uses two additional components: a *Selection Layer* $\mathtt{N_{sl}}$ and a generator $\mathtt{N_G}$ to explore the latent space. Both components' configurations are taken from the best results in the original work Adornetto & Greco (2023). $\mathtt{N_G}$ is implemented for both datasets as a fully connected neural network of 2 layers with $6 \cdot \ell$ and $3 \cdot \ell$ neurons. The dimensionality of $\mathtt{N_{sl}}$ is $k = 30$.

### B.3 METRICS

For the evaluation of our approach we used three metrics, namely, spectral root mean squared error (srmse) and one-hot as defined in Adornetto & Greco (2023), and valid percentage of materials. With the latter metrics, we evaluate the percentage of valid materials over the set of $T$ initialization points. Let $\alpha$ be the number of samples in the test set. We recall that inverse design is performed $T$ times for each element in the test set. Let $\mathcal{V}_i^{\phi}$ be the set of valid materials generated for the $i - th$ element in the test set for a constraint $\phi$.

For a given constraint $\phi$:

$$\mathrm{valid}_{\phi}(\%) = \frac{1}{\alpha} \sum_i^{\alpha} \frac{|\mathcal{V}_i^{\phi}|}{T} \times 100$$

with $\alpha$ number of samples in the test set, and $\mathcal{V}_i^{\phi}$ set of valid materials satisfying $\phi$, out of the $T$ initialization points for the $i - th$ sample in the test set.

## C HARDWARE AND TIMING

We conducted comparative experiments on the time requirement in the same Python environment on Ubuntu 22.04.01, over four 12-core Intel(R) Xeon(R) Gold 5118 CPUs (2.30GHz), 504GB RAM, and two NVIDIA Tesla V100 GPUs (16 GB each). The results are reported in Table 4. It is worth noticing that, while VAE-ID and NA optimize multiple starting points in parallel, GIDNET optimizes the same points sequentially. This causes the inverse design runtime of GIDNET to increment linearly with the number of starting points.

The code was developed in Python 3.12.9 and relies on key libraries such as `PyTorch` (version 2.6.0). A comprehensive list of all packages and their exact versions is provided in the `requirements.txt` file. All the experiments are fully reproducible and random seeds have been properly defined for this purposes in the code. Detailed instructions to reproduce the experiments can be found in the `README.md` file within the code repository, which is included as supplementary material and will be made publicly available upon acceptance.

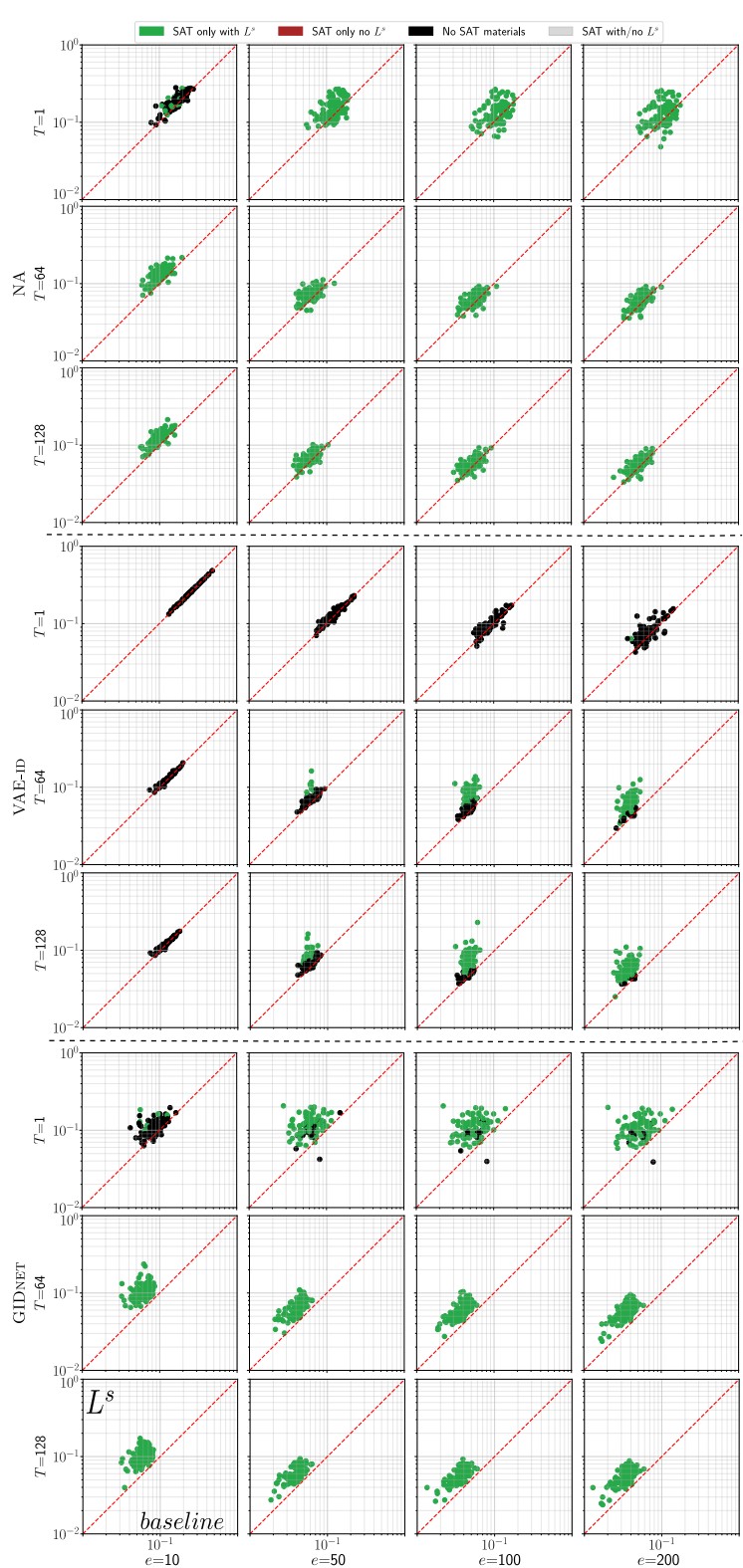

Figure 5: Results for the P4 layout constraint on $\mathcal{D}_{\ell=10}$.

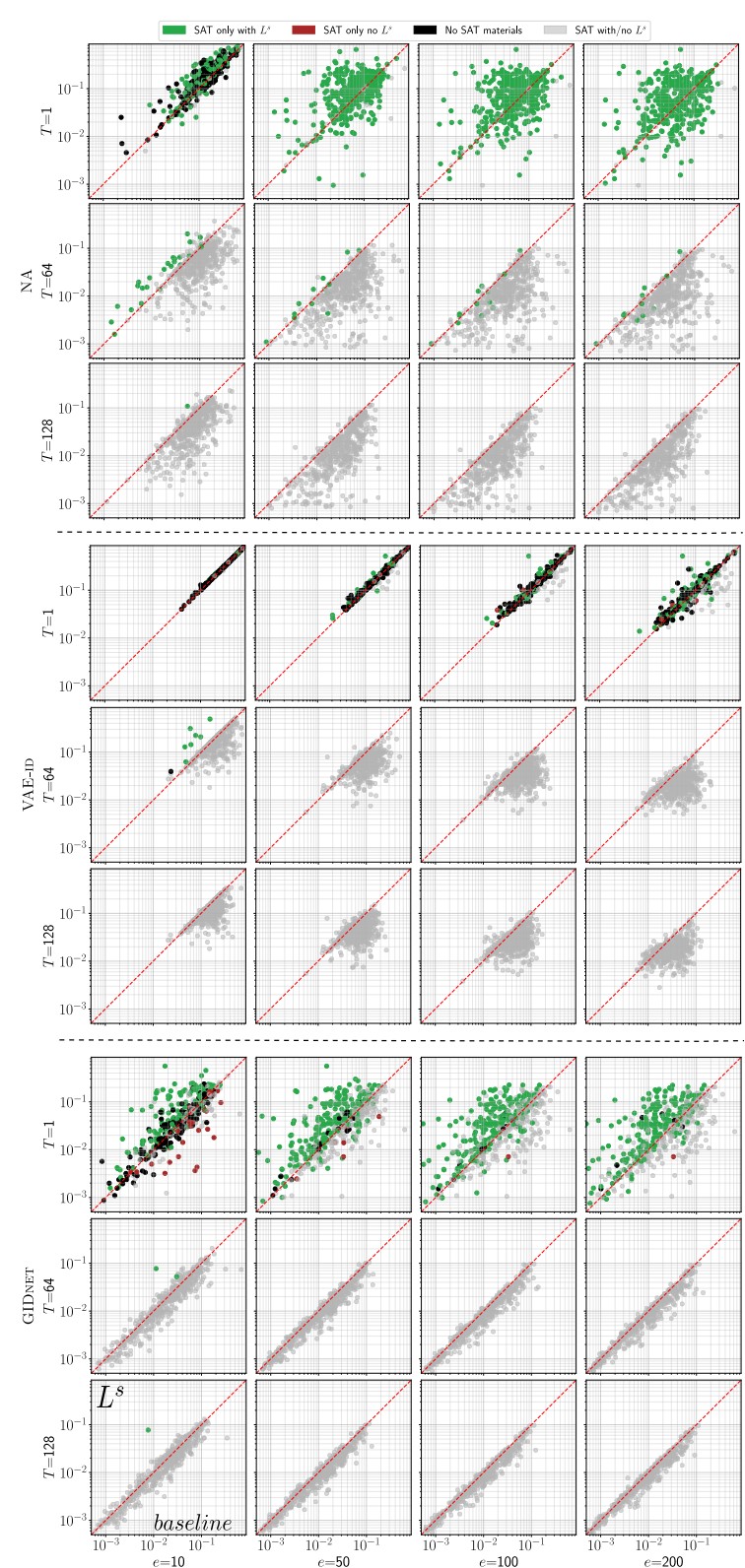

Figure 6: Results for the UA layout constraint on $\mathcal{D}_{\ell=5}$.

## D  ADDITIONAL RESULTS

In the following, we show additional results considering a set of different constraints and different datasets with respect to the ones reported in the main paper.

Figures 5 and 6 replicate the scatterplot layout introduced in the main paper, comparing the SRMSE of materials found by the SL-aumented and baseline methods. The red diagonal marks equal performance on the two approaches.

The former figure shows results for the P4 layout constraint on $\mathcal{D}_{\ell=10}$. All solutions are obtained via SL-augmented optimization at the price of higher SRMSE compared to the baseline, which, nevertheless, produced invalid material. As materials satisfying the P4 layout constraint are absent from $\mathcal{D}_{\ell=10}$, this highlights the contribution of the Semantic Loss in scenarios where the constraint is not represented in the training data.

Figure 6 shows results for the UA layout constraint on $\mathcal{D}_{\ell=5}$. In all the methods, as $T$ and $e$ increase, we can notice that both approaches lead to the discovery of valid materials. Indeed, such valid materials are already well represented in the training set (see Table 2 in the main paper), which contains a notable amount of material that satisfies the UA layout constraint. Thus, it is also probable for baseline methods to produce valid metamaterials. However, the Semantic Loss improves the exploration process, leading to materials with a lower SRMSE than their baseline counterpart, as we can observe from the mass of gray points below the bisector.

Figure 7 presents a series of histograms for the layout constraints PAL2, PAL3 and PAL4 on $\mathcal{D}_{\ell=10}$. In each plot, the number of valid materials found with Semantic Loss optimization is shown in blue, while the baseline (without Semantic Loss) appears in orange. Starting from the first constraint, PAL2, we can observe that the data reflects the previously observed results, where SL-augmented architectures are able to find valid materials in the early stages of exploration, whereas baseline models need more search time and starting points. We also notice how, on average, the mean of the distribution for the SL-augmented models is shifted to the left, towards lower SRMSE values compared to the baseline. When transitioning to the intermediate constraint, PAL3, the difference between the two approaches becomes more pronounced. The SL-augmented models still achieve high numbers of valid materials (also early in the process), in contrast, the baseline performance starts to drop as the design space narrows. The latter layout constraint, PAL4, is the one least represented in the original data (0% constraint satisfaction in both the training and test split of $\mathcal{D}_{\ell=10}$). Nonetheless, the SL-augmented architectures achieve excellent results while the baseline models struggle to find valid solutions. From this disparity, we can draw two conclusions. First, Semantic Loss drives the exploration process of the models towards regions of the design space that satisfy the target layout even when no such examples exist in the training data. Second, the gap between the number of valid materials found by the SL-augmented and baseline widens as the constraint becomes more stringent.

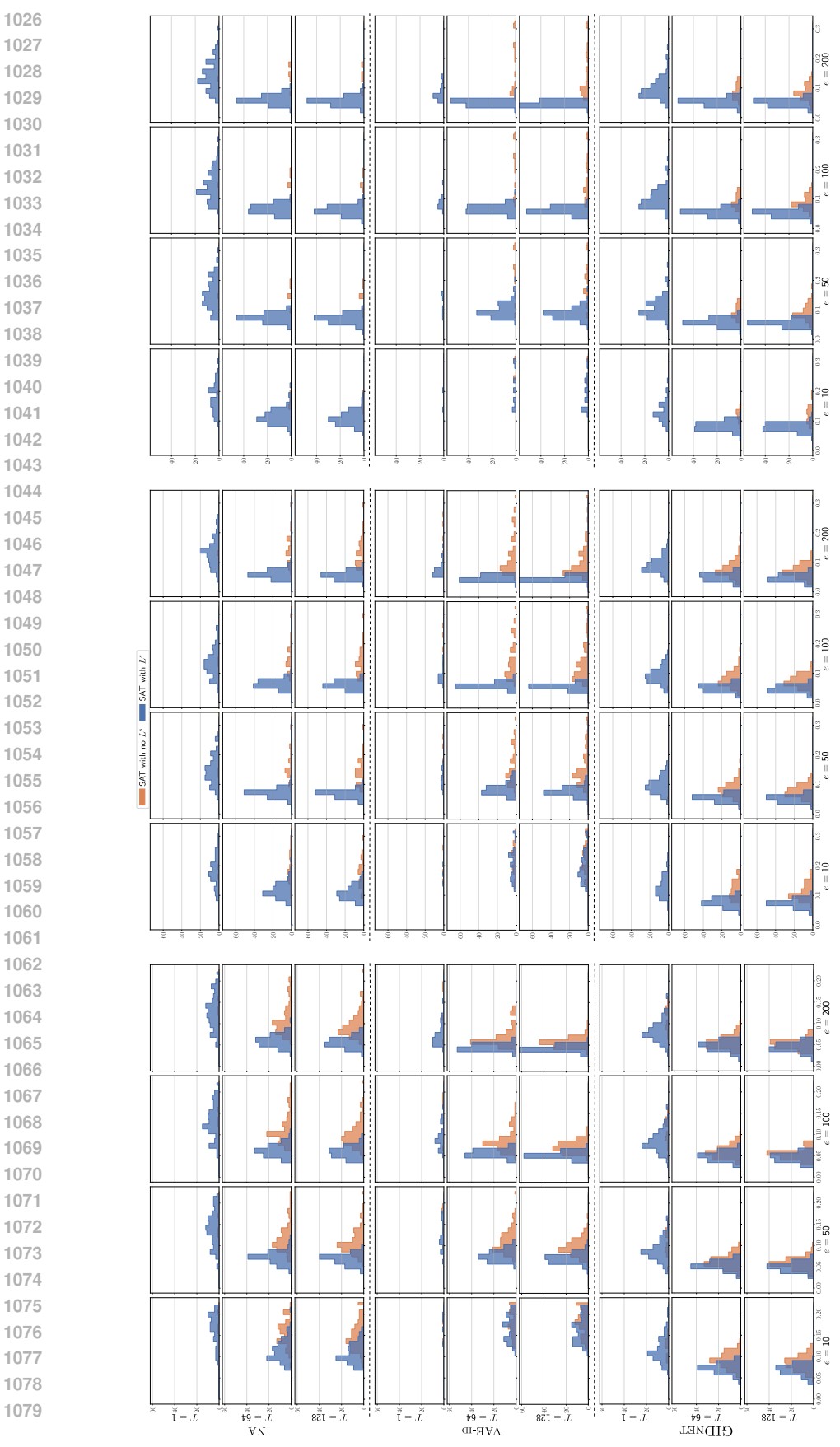

Figure 7: Side-by-side results for the PAL2, PAL3, and PAL4 layout constraints on $\mathcal{D}_{\ell=10}$.

