# OpenReview forum: "A Neuro-symbolic Approach to Inverse Design of Thin-layer Metamaterials Under Layout Constraints"
_ICLR.cc/2026/Conference — ICLR 2026 Conference Withdrawn Submission_

### Official Review · Reviewer_M4YQ · 2025-10-29

**Soundness:** 3
**Presentation:** 2
**Contribution:** 2
**Rating:** 2
**Confidence:** 3

**Summary:**

The paper proposes a neuro-symbolic approach to inverse design of thin-layer metamaterials under layout constraints. The main idea is to augment existing inverse design pipelines (Neural Adjoint, VAE-ID, GIDNET) with a differentiable logical penalty called Semantic Loss.

Each layer’s material choice is treated as a discrete variable, and user-specified structural constraints, such as no adjacent duplicates, palindromic stacks, fixed periodic repetition, use all materials at least once, are encoded as a Boolean formula $\phi$.

The Semantic Loss is defined as the negative log-probability that the model’s soft assignment over materials would produce a layout satisfying $\phi$. During inverse design, the method jointly optimizes optical performance via SRMSE between predicted and target spectra and this Semantic Loss, so that resulting structures both match the target spectrum and satisfy fabrication-style layout rules.

On two synthetic multilayer datasets 5-layer / 10-layer, adding Semantic Loss substantially increases the percentage of valid designs under hard constraints and often approaching 100% vs. <30% for baselines while keeping spectral error comparable.

Overall, while I see the improved results with the loss term on specific task, I do not think this paper has strong technical contribution or solid validation to make it accepted to ICLR.

**Strengths:**

- Addresses a practically important problem in photonic / metamaterial design: enforcing discrete structural/layout constraints that matter for manufacturability.

- Uses a general logical formalism: constraints are expressed symbolically and turned into a differentiable loss without retraining the forward surrogate as previous research do.

- Empirical results show clear gains in constraint satisfaction, including on constraints that are extremely rare or entirely absent in the training data. Valid design rate jumps from very low <~30% to near 100% while keeping SRMSE roughly stable.

- Tested across multiple inverse design backbones, e.g., Neural Adjoint, VAE-ID, GIDNET, suggesting the idea is not tied to a single architecture.

**Weaknesses:**

- ML novelty isn’t clear enough for ICLR.
The main technical piece (semantic / neuro-symbolic loss that enforces logical constraints) already exists in prior neuro-symbolic work. Here it’s mainly being applied to multilayer optical design. Right now this reads more like an application of an existing idea than a new ML method.
- The actual objective being optimized is not written down cleanly.
It seems like you’re minimizing SRMSE + λ * SemanticLoss, but the paper never explicitly gives that full loss, never defines λ, and never explains how λ is chosen. That hurts clarity and reproducibility.
- No sensitivity / ablation on the constraint term.
The authors claim you can enforce strict constraints “without hurting performance,” but we only see one operating point. There’s no sweep of λ to show the tradeoff between spectrum error vs % valid designs.
- Missing a strong baseline.
The authors compare baseline vs baseline+SemanticLoss, but not against a very natural alternative: run normal inverse design for many trials, then post-process and repair the layout to satisfy the constraint and fine-tune thickness. Without that baseline, it’s hard to prove the in-loop semantic loss is necessary.
- All demonstrations are on simulated thin-film stacks with 5 or 10 layers. The narrative promise is broader (“a general neuro-symbolic approach for inverse design under layout constraints”), but there is no second domain (e.g. metasurface phase masks, RF stacks, etc.), no fabrication/measurement validation, and no demonstration that the same pipeline transfers to a qualitatively different physical system.
For a top ML venue, a purely single-domain, simulation-only study with limited ablations would not constitutes strong contribution.
The authors need to either add more physical design task with different constraints, to argue this is a general constrained-inverse-design recipe, or fabricate & measure designed structures and show that satisfying the symbolic constraints is not just cosmetic but tied to real manufacturability and performance. Either route would convince more on AI for Science with demonstrated impact.
- The authors claim the semantic loss finds valid designs faster and earlier in the search, but I don’t see convergence curves or time-to-valid plots.

**Questions:**

- What is the exact objective the authors optimize in the paper?
  Please write the full loss minimized during inverse design, including all terms and coefficients.
  If the obj is of the form
  $\mathcal{L} = \text{SRMSE} + \lambda \cdot L_s(\varphi, \bar{M})\$,
  please confirm that and define every symbol.

- How is $\lambda$ chosen?
  Is $\lambda$ fixed across all constraints and all architectures, or tuned per setting?
  If tuned, how? If fixed, why does that work?

- Can the authors show any tradeoff curve?
  Can the authors sweep $\lambda$ and plot % valid designs vs SRMSE?
  This would directly support the claim that one can satisfy hard constraints without sacrificing optical performance.

- Why wouldn’t a simple post-hoc repair baseline work just as well?

- How general is the approach beyond multilayer thin-film stacks?
  Does same semantic-loss machinery to work on a different physical inverse design task (e.g. metasurface phase masks, RF stacks, etc.) without major changes?

- Can the authors quantify "“faster to find valid designs"?
  The authors state that adding the semantic loss helps the optimizer reach valid designs earlier while convergence plot are needed in rebuttal , e.g. probability of obtaining a valid design vs. iteration/restart index.

- The authors need to further clarify the novelty claim for general ML audience.
My current take is that the neuro-symbolic loss itself is not new as it follows prior work on semantic loss for logical constraints, and the main contribution here is showing how to plug that loss into an inverse photonic design loop and get high-validity designs for constraints that never appeared in training data.

---

### Official Review · Reviewer_R1AF · 2025-10-31

**Soundness:** 2
**Presentation:** 3
**Contribution:** 2
**Rating:** 2
**Confidence:** 4

**Summary:**

The paper proposes a way to convert layout constraints in thin-film metamaterial inverse design into differentiable losses, specifically, a semantic loss that quantifies how well the designed metamaterial satisfies the layout constraints. The author demonstrates the effectiveness of this loss by applying it to three different benchmark models and two different layer numbers, showcasing an increase in valid designs.

**Strengths:**

The authors provide a good amount of validation on different designs.

**Weaknesses:**

It seems that the core idea is to augment standard inverse design pipelines with a differentiable logical constraint term (the semantic loss). Comparison with other methods to apply constraints besides the baseline is limited.

**Questions:**

1. How is the semantic loss applied on top of the original loss (the difference between the output spectral response and the target spectral response of the material, which is also not explicitly written down in the paper)? Is there a hyperparameter that weighs between the two losses? If so, how is that parameter chosen, and how does that affect the performance of the method?
2. The current constraints only reflect on the binary matrix M, which decides the layer material. How to apply layer thickness constraints?
3. There are more straightforward ways to apply those constraints during optimization. For example, instead of treating every element in M as a free variable, you can 1) assign layer i and layer L-i to be the same variable to satisfy PALINDROME constraint; 2) assign layer i and layer i+k to be the same variable to satisfy PERIODICAL constraint; 3) add a loss between neighboring layer variables to force them to be different materials. Have the authors tried these, and how would these approaches compare with this method?
4. Intuitively, running a genetic algorithm that uses the learned surrogate model for evaluation and discards the sampled structure conflicting with the layout constraints would also yield a comparable performance. The author should compare their method with this in terms of convergence speed and material property accuracy.
5. How does this method generalize beyond this very specific layer-based metamaterial design? The value of this approach would be very limited if it is just crafting a new loss term with targeted applications.

---

### Official Review · Reviewer_egDE · 2025-11-03

**Soundness:** 2
**Presentation:** 2
**Contribution:** 1
**Rating:** 2
**Confidence:** 5

**Summary:**

This paper proposes using semantic loss (SL) to enforce constraints during the inverse design of optical metamaterials. However, the value and novelty of the proposed methodology are not clearly articulated. The paper is ambiguous regarding key aspects of the proposed framework, and the training and inverse design framework diagrams are not provided. Furthermore, the case studies presented use input constraints that appear trivial.

**Strengths:**

The proposed method appears to be scalable to different numbers of constraints.

**Weaknesses:**

1. This paper is not well motivated. In the introduction and the literature review part of the paper, the author's solely focus on generative methods for inverse design, however they fail to motivate as to why this problem even needs generative methods. The paper only addresses input constraints during inverse design, which are trivial to handle as they do not require evaluating an expensive objective function (i.e., the frequency spectrum) and there is already extensive literature on handling such constraints in the design optimization domain. The other potential source of complexity involves discrete material variables; however, gradient-free optimizers such as binary genetic algorithms (GAs) are well suited to such problems, especially when surrogate models are used, but no comparisons to such methods are provided.

2. The SRMSE metric may not be appropriate for inverse design of metamaterials, as the primary goal is typically to match the frequency response in specific regions (usually to search for bandgaps). Therefore, comparing the entire frequency spectrum can be meaningless in this context.

3. Adding to this, the general ambiguity of the paper is a major weakness. It is not even clear where the proposed loss is used, whether it is applied to train the surrogate model or as an additional objective function during the “inverse computation” process. The SRMSE loss, which appears to be an RMSE loss applied across multiple dimensions, is not included in the paper. Instead, the authors provide only a secondary reference, which further obscures the work.

**Questions:**

1. The authors need to provide a clear justification for using generative model–based methods for this problem in the first place. They should explicitly identify the limitations of existing state-of-the-art methods and then outline how their proposed framework addresses these issues. In addition, the choice of loss function requires justification. For example, in cases where consecutive spots should not have the same material, the proposed loss may be unnecessary, simply using loss = $ -\Sigma_{i=0}^{n}(m_j - m_i)^2 $   (capped at a lower bound) would be sufficient. How does the proposed semantic loss benefit here?


2. Given the amount of training data and the use of surrogate models, it seems plausible that a conventional surrogate-based optimization, possibly even with gradient-based optimizers when warm-started with the most optimal sample from the training data would perform comparably or better (without additional training costs). The authors need to add these baselines to the results in addition to listing the computational costs of training the generative models.

3. The entire inverse design pipeline is unclear, the authors need to add a framework diagram to explain it clearly. It is unclear whether the proposed SL loss is used as a loss function for the inverse design process or for training the surrogate model. If it is the latter, this should be explicitly stated. Additionally, important implementation details such as the choice of optimizer should be clearly stated.

4. The authors should specify the dimensionality of the design problem. How many cells are included in each metamaterial configuration, and into how many frequency points is the frequency spectrum discretized? These details are essential for understanding the problem’s overall complexity and assessing the feasibility of the proposed approach.

---

### Note · Authors · 2025-11-28

I have read and agree with the venue's withdrawal policy on behalf of myself and my co-authors.